# Doppler Ultrasound Selection and Follow-Up of the Internal Mammary Artery as Coronary Graft

**DOI:** 10.3390/biomedicines11010066

**Published:** 2022-12-27

**Authors:** Pietro Scicchitano, Micaela De Palo, Giuseppe Parisi, Margherita Ilaria Gioia, Marco Matteo Ciccone

**Affiliations:** 1Department of Cardiology, Hospital ‘F. Perinei’ Altamura, 70022 Altamura, Italy; 2Department of Cardiac Surgery, University ‘A. Moro’ Bari, 70124 Bari, Italy; 3Department of Cardiology, Hospital ‘S. Paolo’ Bari, 70124 Bari, Italy; 4Department of Cardiology, Hospital ‘Perrino’ Brindisi, 72100 Brindisi, Italy; 5Department of Cardiology, University ‘A. Moro’ Bari, 70124 Bari, Italy

**Keywords:** coronary artery by-pass graft, Doppler ultrasound evaluation, pulsed wave doppler, continuous wave doppler, stenosis

## Abstract

The impact of coronary artery disease (CAD) on all-cause mortality and overall disabilities is well-established. Percutaneous and/or surgical coronary revascularization procedures dramatically reduced the occurrence of adverse cardiovascular events in patients suffering from atherosclerosis. Specifically, guidelines from the European Society of Cardiology on the management of myocardial revascularization promoted coronary artery by-pass graft (CABG) intervention in patients with specific alterations in the coronary tree due to the higher beneficial effects of this procedure as compared to the percutaneous one. The left internal mammary artery (LIMA) is one of the best-performing vessels in CABG procedures due to its location and its own structural characteristics. Nevertheless, the non-invasive assessment of its patency is challenging. Doppler ultrasonography (DU) might perform as a reliable technique for the non-invasive evaluation of the patency of LIMA. Data from the literature revealed that DU may detect severe (>70%) stenosis of the LIMA graft. In this case, pulsed-wave Doppler might show peak diastolic velocity/peak systolic velocity < 0.5 and diastolic fraction < 50%. A stress test might also be adopted for the evaluation of patency of LIMA through DU. The aim of this narrative review is to evaluate the impact of DU on the evaluation of the patency of LIMA graft in patients who undergo follow-up after CABG intervention.

## 1. Introduction

Coronary heart disease (CHD) is one of the most common causes of death and disability in developed countries [1]. More than one-third of patients continue to die in Western countries due to CHD [2,3]. Data from the 2016 Heart Disease and Stroke Statistics update of the American Heart Association (AHA) estimated that CHD prevalence is increasing, while myocardial infarction (MI) seems to occur approximately every 42 s in American population [4]. European data reveal that cardiovascular disease (CVD) accounts for more than 4 million deaths each year, while no differences were between men and women [5].

Coronary angiography (CA) is the gold standard technique for detecting coronary atherosclerotic plaques, which are responsible for the occurrence of acute coronary syndromes (ACS) or stable angina. The results from CA imaging led physicians to choose between percutaneous or surgical revascularization for saving myocardial viability and preventing CHD comorbidities and mortality.

ESC guidelines on myocardial revascularization [6] provide the indications for the use of coronary artery by-pass graft (CABG) rather than percutaneous procedure: patients with two-vessel disease with proximal left anterior descending (LAD) coronary artery stenosis (level of evidence IB), left main disease independently from SYNTAX score (level of evidence IA), three-vessel disease independently from SYNTAX score values (level of evidence IA), and other peculiar situations should be considered for surgical revascularization [6].

The left internal mammary artery (LIMA) is the graft of choice for revascularization of the left anterior descending coronary artery (LAD) [7,8,9] because of its long-term patency rate [10].

The functional evaluation of the coronary graft is crucial for risk stratification of patients who underwent CABG. Despite the reduced rate of occlusion, the probability of a progressive narrowing process due to atherosclerosis progression remains high enough to require a comprehensive diagnostic evaluation of CABG patency. The conventional and current gold standard investigation for the assessment of conduit stenosis is CA, but its invasiveness includes many risks such as vascular trauma, arrhythmias, stroke, myocardial infarction, and arterial dissection.

Different methods that directly and non-invasively analyze the patency of the graft have been recently developed. Multidetector computed tomography (CT) showed good sensitivity and specificity [11], but it is expensive and exposes patients to radiation risk [12]. Magnetic resonance imaging (MRI) is an alternative to CT, but it does not yet provide sufficient sensitivity and specificity for the assessment of the function of coronary grafts [13].

Doppler ultrasonography (DU) can be considered as a further non-invasive tool for detecting LIMA patency after CABG as firstly reported by Fusejima et al. [14]. DU has shown promising results in studying the aorto-coronary graft and, specifically, the function of the LIMA, even if only its proximal portion can be visualized.

The aim of this review is to evaluate the role of DU in assessing LIMA graft patency during the follow up of patients with CHD treated with CABG.

## 2. Anatomy of the Left internal Mammary Artery (LIMA) and Coronary Bypass Graft

LIMA arises from the subclavian artery opposite to the thyreocervical trunk and descends behind the cartilages of the upper six ribs at a distance of about 1.5 cm from the margin of the sternum. Two mammary veins whose length vary from 15.1 to 26.0 cm run parallel to LIMA [15,16].

Other arterial conduits can be adopted for myocardial revascularization purposes: right internal mammary artery (RIMA); radial artery; right gastroepiploic artery; and, rarely, ulnar artery and/or inferior epigastric artery [17].

All of these are medium-calibre arterial vessels (diameter 2.5–5 mm). The large amount in elastic components, the internal and external elastic membranes, and the endothelial layer—which is able to act as a paracrine and autocrine organ—are all suitable features for their use in CABG. Due to their structural and functional characteristics, they are able to actively modify their lumen, regulate the amount of blood flowing to organs and tissues, and preserve their own structure from damages deriving from manipulation during the procedure [17].

LIMA is the best vessel to choice for the revascularization of occluded coronary arteries—LAD in particular—due to its location and structural characteristics.

Despite the possible use of RIMA, the most widely used graft is the left counterpart: the position of the LIMA in mediastinum, its route and proximity to the heart, and the possibility for an easy surgical anastomosis of the LIMA—as compared to RIMA—to the coronary artery are the most important reasons for routinely adopting LIMA rather than RIMA [18].

LIMA is recognized by cardiothoracic surgeons as the most effective and reliable conduit due to its structural characteristics, lower incidence of adverse events, and greater long-term patency rate [19,20,21,22,23]. The histological characteristics of this conduit are effectively responsible for the lower incidence of adverse events as they confer the vessels relative freedom from atherosclerosis as well as from intimal hyperplasia [24,25].

LIMA presents histo-morphological characteristics of an elastic artery as it is possible to recognize the intima, the media, and the adventitia [26,27]. Nevertheless, the composition of these layers may vary throughout the route of LIMA from the subclavian artery to the epigastric bifurcation. The major amount of elastic fibers in tunica media is found in the middle part of the internal mammary artery course as the distal one shows a few elastic fibers [17]. Furthermore, previous studies outlined the higher rate of elastic fibers in the LIMA structure as compared to the RIMA [17,18]. The tunica media is surrounded by two elastic lamina: the external one, which gives the most contribution to the elastic performance of the vessel, and the inner lamina, which is mainly composed of muscular fibers [17,18,24]. All of these structural characteristics account for the patency properties of the LIMA and its usefulness in the case of CABG intervention. Furthermore, LIMA showed fewer endothelial fenestrations and less intercellular junction permeability, as well as higher intercellular communication: such characteristics provide high resistance to LIMA during the manipulation when harvested for CABG [28].

During the intervention, if not performed off-pump, the patient’s aorta and heart are centrally cannulated, and attached to the cardiopulmonary bypass (CPB) circuit. Then, after placing an occlusive cross-clamp on the ascending aorta, the heart is arrested using a high-potassium concentration cardioplegic solution. Then, the surgeon can anastomize the harvested conduit to the coronary artery distal to the zone where the flow is blocked. Then, the cross-clamp is removed and the cardioplegia washed out, the heart begins to contract, and the surgeon can check the grafts for both blood flow and competence as well as possible bleeding from the anastomosis site.

Therefore, the resistance of vessels to manipulations should be really high in order to prevent early and late stenosis of the graft. The morphological and ultrastructural characteristics of the vessels account for this resistance. Such resistance is higher than saphenous grafts, whose manipulation can easily lead to their early occlusion soon after CABG intervention [28].

## 3. Non-Invasive Assessment of Coronary Arteries and Grafts: The Role of Echo Color Doppler

The echo-color Doppler technique represents one of the most attractive instruments for the evaluation of coronary grafts. Despite some limitations related to the reproducibility of the technique, its performance by experts constitutes one of the best methods for the non-invasive assessment of graft patency.

### 3.1. Color Doppler Ultrasonography Technique

The non-invasive approach to the evaluation of LIMA based on echo-color Doppler is complex and challenging.

Linear probes are the first line instrument in the evaluation of these vessels, followed by the pencil beam, which allows for a continuous-wave Doppler signal that is able to provide useful information about the hemodynamics. Specifically, pencil probes are able to recognize blood flow by means of a transducer—the Pedoff’s transducer—whose shape is like a pencil and is formed by two separated elements: one that is able to transmit the signal, and the one that is able to receive the backward signal.

The linear probe is a 2–4 MHz transducer directly placed on the skin after application of commercial ultrasonic gel. The LIMA graft can be visualized either from the left supraclavicular fossa [29] or from the left parasternal window of the intercostal spaces [30].

#### 3.1.1. Supraclavicular Window

The left supraclavicular fossa approach is the preferred one. To detect the LIMA flow, the initial velocity range of the color Doppler should be set at approximately 21 cm/s and then adjusted to optimize the visualization of color signals in the bypass graft [31]. The proximal LIMA is detected by slightly rotating the transducer clockwise, then inclining it toward the anterior chest wall. The flow velocity in LIMA is studied about 2.0 cm far from the left subclavian artery by pulsed-wave Doppler echocardiography with a sample volume of 2 mm after correcting the flow angle.

#### 3.1.2. Left Parasternal Window

The possibility for detecting LIMA flow is lower when the parasternal approach is adopted [32]. Indeed, a modified left parasternal window can be used to detect LIMA with the patient lying in the left lateral or supine position. The vessel is parasternally located within the second to fifth left intercostal space. The LIMA graft is identified as a tubular structure with color flow directed from the base to the apex. Intraluminal flow signals are obtained using the pulsed-wave Doppler method after identifying the LIMA position [33].

### 3.2. Coronary Circulation Physiopathology and Doppler Flow Characteristics

Preoperative supraclavicular and transthoracic ultrasound of the LIMA shows higher systolic and lower diastolic velocity patterns as in peripheral arteries. Normal LIMA arterial Doppler is usually characterized by a biphasic or triphasic signal [34]. The first component—defined as peak systolic velocity (PSV)—corresponds to the higher velocity of the forward flow during systole. The second is a lower frequency signal named end-diastolic velocity (EDV); it is a reversed flow that occurs during early diastole. The third component of the pulsed-wave Doppler signal is the lowest frequency and represents forward flow in late diastole (PDV).

The CABG intervention changes the Doppler profile of the LIMA: the flow in proximal LIMA becomes biphasic (systolic and diastolic). It occurs as a sort of “diastolization” of LIMA blood flow: the Doppler shift frequency drops during the systolic phase and appears more prominent during the diastolic phase [35,36]. The diastolic flow velocity of the LIMA increase is the result of the physiologically decreased resistance in the coronary circulation. Whilst the distal LIMA adapts itself to the coronary artery flow dynamics (i.e., predominantly diastolic flow), proximal LIMA, i.e., the portion near the origin from the subclavian artery, will predominantly retain a systolic flow.

## 4. LIMA Graft Patency: Postoperative Assessment

The prognosis of CABG patients depends on the demonstration of the aorto-coronary by-pass graft functionality. Routine postoperative CA could not be performed in relation to its intrinsic risks.

Doppler ultrasound (DU) was developed as an alternative to CA, but it is rarely used due to its suboptimal sensitivity and specificity. Improvements in technological resources and technique have been obtained over the years.

Anastomosis evaluation is challenging because blood flows from both the native LAD and LIMA to the anastomotic site, thus causing competitive flow [37].

Difficulties in blood flow detection are also related to the evaluation of the stenosis degree. Elevated flow velocity at DU represents true stenosis; nevertheless, moderate-to-severe stenosis (50–70%) of the LIMA graft is hardly detected by means of DU as flow does not significantly differ from data coming from less-occluded grafts [37].

The pulsed wave (PW) Doppler is the best DU method for the non-invasive assessment of coronary graft stenosis. The first characteristic to be considered is the morphology of the Doppler spectrum profile. Proximal occlusion of the LIMA graft usually makes the flow monophasic, thus showing the loss of the diastolic component; distal stenosis can determine the decrease in diastolic flow pattern, although the biphasic pattern is preserved. Graft failure could be suspected when no flow or a systolic dominant pattern with reduction in diastolic component is detected [38].

Nevertheless, further parameters should be considered when using PW Doppler in order to better identify severe (>75%) stenosis of the LIMA graft. In particular, the physicians’ attention should be concentrated on the following parameters (Figure 1):

The peak of the systolic velocity (PSV): a predominant PSV at baseline is considered as a marker of LIMA graft dysfunction [39].

The peak of diastolic velocity (PDV): the higher the baseline values of PDV, the lower the risk for graft stenosis.

PDV/PSV ratio: due to its independence from the angle of incidence of the ultrasound beam, it is a reliable parameter to be considered for the evaluation of the LIMA graft. Bach et al. [37] observed the increase in the PDV/PSV ratio from 0.6 (proximal edge) to 1.4 (distal edge). The higher the ratio, the lower the probability of a significant LIMA graft stenosis [37]. A decrease in PDV/PSV ratio (<0.5 or 0.6) is predictive of graft severe stenosis, as several studies already demonstrated [40]. A DPV/SPV ratio greater than 1 is associated with good angiographic findings and can provide sensitivity up to 100% and specificity proximal to 58% for graft patency detection [41,42].

Diastolic fraction (DF): it expresses the impact of the diastolic flow on the composite velocity time integral (VTI, express in meters) of both systolic and diastolic flow. The diastolic VTI fraction is obtained by dividing the diastolic VTI (DVTI) by the sum of the systolic (SVIT) and DVTI according to the following formula [33,43]:DF = DVTI/(DVTI + SVTI)(1)

The DF better performs than DVTI as it seems to overcome the intrinsic limitation of the latter. Specifically, reduced inter- and intraobserver variability can be obtained by dividing DVTI to the sum of DVTI plus SVTI. Leitão et al. [43] found good reproducibility in detecting LIMA graft patency, with the cut-off being set at <50% for the detection of significantly hemodynamic stenosis (i.e., >70% reduction in lumen diameter). These data are corroborated by those from Takagi et al. [44], who compared the duplex Doppler echocardiography to the Doppler catheter measurement of the LIMA graft flow. Despite the small sample size, the analysis revealed a higher correlation between DF and invasive measurements (r = 0.90, *p* < 0.01), as well as other single Doppler parameters (PSV: r = 0.84, *p* < 0.01; PDV: r = 0.79, *p* < 0.01, with a diastolic to systolic peak flow velocity ratio: r = 0.86, *p* < 0.01) [44].

Beyond the Doppler parameters at rest, pharmacological tests can be further adopted to evaluate the LIMA graft patency and corroborate the data at rest. The changes in flow velocity after the administration of coronary vasodilator drugs such as dipyridamole (which acts on resistance vessels) or nytroglicerin (NTG, which acts on capacitance vessels) can provide useful, dynamic information regarding the evaluation of the LIMA graft. Both of them are able to increase PDV without influencing PSV. In particular, dipyridamole infusion promotes stable blood pressure while increasing heart rate; therefore, the diameter of the LIMA-graft tends to increase, while flow velocity remains stable, thus providing an increase in vessels capacity. NTG administration causes the increase in PDV in LIMA, with no modifications in systolic peak velocity and diameter. Generally, the increase in oxygen demand during DU stress test will cause an increase in the velocity of the flow into the graft: this will confirm the patency of the graft itself [45,46].

## 5. The Continuous Doppler Evaluation

The continuous-wave (CW) Doppler evaluation of the blood flow velocity in the LIMA graft is a further, easy, and reproducible technique for evaluating human coronary flow [47,48].

Pencil probes using 8 MHz frequency are the instruments to be considered for the recording of the CW Doppler images of the LIMA graft.

### Doppler Evaluation

After positioning the patient supine on the bed in a quiet, temperature-controlled room, the sonographer could stand behind his/her head in a comfortable resting position, with the elbow preferentially put stable on the bed in order to maintain a stable position.

The probe should be positioned at the left supraclavicular fossa with an angle manually maintained at 75 to 80 degrees: slow movements should be attained by changing the incidence angle of the probe in order to correctly identify the LIMA graft blood flow.

The normal, not-anastomized LIMA typically shows a CW pattern characterized by higher systolic and minimal diastolic waves (Figure 2).

The reason is related to the physiology of the vascular cycle: peripheral vascular beds are normally perfused during the systole, thus accounting for the higher peak of systolic Doppler profile as compared to diastolic one in vessels still not-anastomized to the coronary arteries. After the by-pass intervention, the situation of perfusion changes. The coronary artery is usually perfused during diastole. Therefore, the LIMA can show a new, double situation after its connection to the LAD (Figure 2): it can maintain the systolic peak, but a more pronounced diastolic peak can be observed if the graft perfusion is not compromised [47,48,49].

This technique is safe and reproducible above all when used for the evaluation of coronary graft anastomized to LAD, as already demonstrated in the literature [48,49,50,51].

Nevertheless, the influence of different conditions on graft flow should be taken into account in order to have a complete overview of the patency avoiding confounding factors. Ciccone et al. [49] analyzed blood velocity curves obtained in normal and anastomized internal mammary arteries during hyperventilation and Valsalva manoeuvre. They found that during hyperventilation, blood velocity increased in the normal mammary but not in the anastomized artery, while during the expiratory effort of the Valsalva manoeuvre, the mean blood velocity decreased in the normal mammary artery but did not change significantly in the anastomized artery [49,52].

## 6. Clinical Application, Limitations, and Future Perspectives

The use of Doppler ultrasound evaluation of LIMA patency represents a reliable opportunity for clinicians for the identification of patients at risk for occlusion of the graft.

Although far from being considered as a definite recommendation, we would like to suggest a dedicated flow chart for the application of the ultrasound evaluation of the LIMA in daily clinical practice. Figure 3 tried to provide advice for the implementation of this technique by suggesting the opportunities to use Doppler ultrasound for short, medium, and long-term follow-up of patients who had been discharged from Cardiac Surgery Unit after CABG.

Specifically, the link between ultrasound data (patency, narrowing, or normal vessel) and the clinical characteristics of patients (symptoms angina-like, chest pain, dyspnea during efforts, etc.) will promote better comprehension of the patients’ needs and the management of them.

Nevertheless, limitations should be considered when applying Doppler ultrasound to the evaluation of LIMA. Firstly, the need for the performance of the examination by an expert might limit the widespread use of the procedure. Secondly, inter- and intraobserver variabilities should be always taken into account when dealing with ultrasound techniques. Thirdly, the reproducibility of this technique should be compared with standard-of-care in dedicated clinical trials. Specifically, there is no study dealing with the comparison between duplex ultrasound and CT angiography for the evaluation of the patency of LIMA or RIMA in patients who had undergone CABG: dedicated clinical evaluations should be finalized for this purpose.

Finally, multimodality imaging techniques should represent the best approach to patients with cardiovascular diseases—and those who underwent CABG in particular—rather than a single technique approach as this will allow for the correct identification of alteration in patients.

## 7. Conclusions

LIMA is the conduit of choice for bypass graft in the revascularization of damaged cardiac muscle after myocardial infarction due to coronary atherosclerosis. Its functional evaluation is crucial during the follow-up of patients who underwent CABG intervention. The gold standard investigation for the assessment of conduit stenosis remains CA, but this is an invasive technique and it entails many risks.

Different non-invasive methods had been recently developed for coronary graft evaluation. Nevertheless, CT and MRI still show limitations. DU seems to be the best non-invasive tool to assess LIMA patency after CABG both immediately and/or late from the intervention. Its good reproducibility, ease of use, and significant correlation with angiographic results make DU the best technique to be adopted in clinical ambulatory practice in order to follow-up with patients who underwent CABG.

## Figures and Tables

**Figure 1 biomedicines-11-00066-f001:**
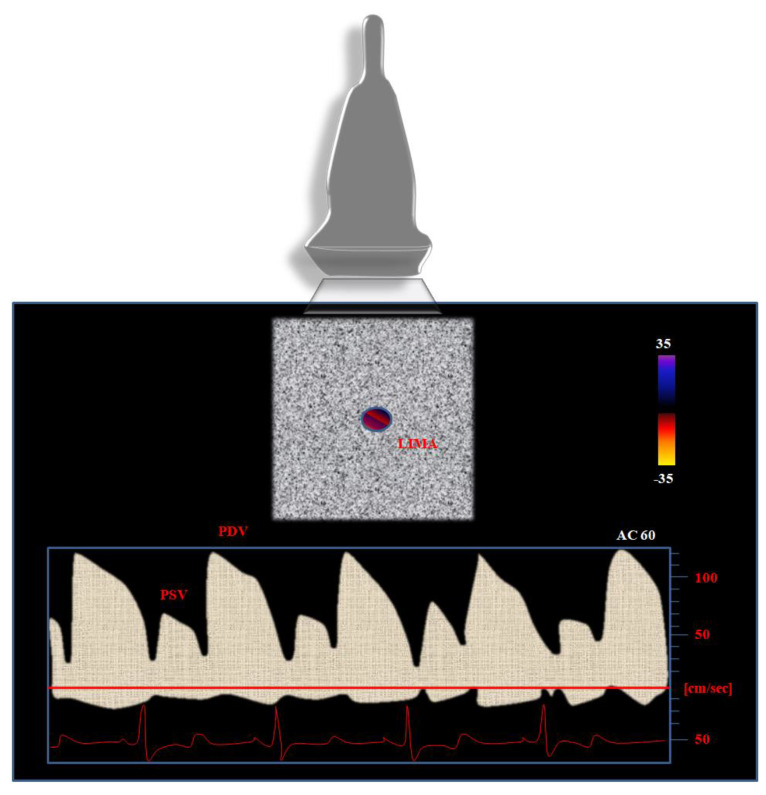
Left internal mammary artery (LIMA) pulsed-wave color Doppler flow aspect from the left parasternal window: the Doppler profile points out the peak systolic (PSV) and diastolic (PDV) velocities.

**Figure 2 biomedicines-11-00066-f002:**
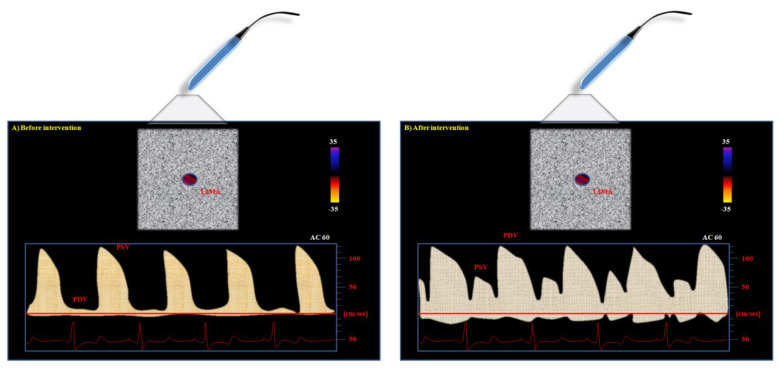
Left internal mammary artery (LIMA) continuous-wave color Doppler flow aspect from the supraclavicular window: (**A**) before aorto-coronary bypass graft (CABG) intervention, and (**B**) after CABG intervention. The differences in Doppler profile (i.e., peak systolic (PSV) and diastolic (PDV) velocities) can be observed.

**Figure 3 biomedicines-11-00066-f003:**
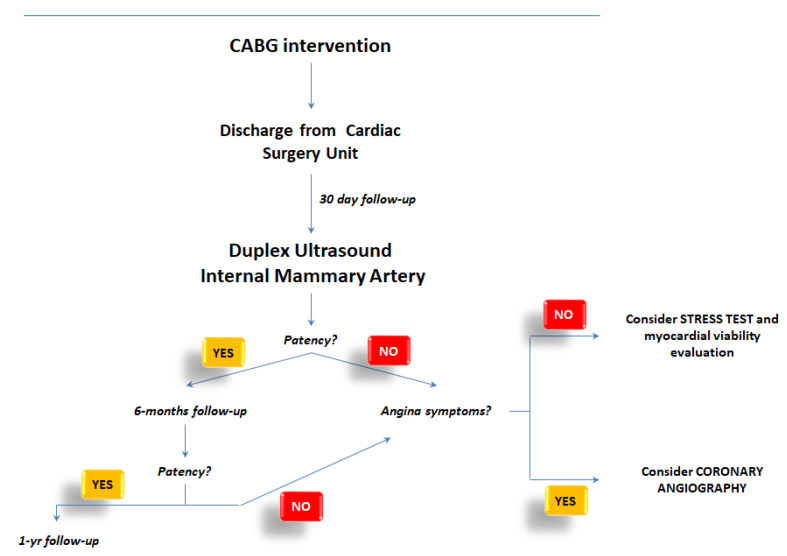
Flow chart for the application of the ultrasound evaluation of the LIMA in daily clinical practice.

## Data Availability

Not applicable.

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
