# Peer review of "Doppler Ultrasound Selection and Follow-Up of the Internal Mammary Artery as Coronary Graft"

_biomedicines, 2022, doi:10.3390/biomedicines11010066_

Round 1
Reviewer 1 Report
The authors aimed to evaluate the impact of Doppler Ultrasonography (DU) on the evaluation of the patency of Left internal mammary artery (LIMA) graft in patients who undergo follow-up after CABG intervention in a narrative review. They concluded that different non-invasive methods had been recently developed for coronary graft evaluation. Nevertheless, CT and MRI still show limitations. DU seems the best non-invasive tool to assess LIMA patency after CABG both immediately and/or late from the intervention. Its good reproducibility, easy way to perform, and its great correlation with angiographic results make DU the best technique to be adopted in clinical ambulatory in order to follow-up patients who underwent CABG.
This is a well-written, informative and comprehensive review.
Comments
1. A figure about the anatomy of coronary arteries would be useful.
2. An algorithm that shows the position of DU in the diagnostic process would be informative.
3. A section about the limitations and insufficiencies of DU should be added.
Author Response
Reviewer #1
We thank this Reviewer for the constructive comments and suggestions. Furthermore, we would like to really thank him/her for his/her appreciation about our research. This is our point to point reply.
- The authors aimed to evaluate the impact of Doppler Ultrasonography (DU) on the evaluation of the patency of Left internal mammary artery (LIMA) graft in patients who undergo follow-up after CABG intervention in a narrative review. They concluded that different non-invasive methods had been recently developed for coronary graft evaluation. Nevertheless, CT and MRI still show limitations. DU seems the best non-invasive tool to assess LIMA patency after CABG both immediately and/or late from the intervention. Its good reproducibility, easy way to perform, and its great correlation with angiographic results make DU the best technique to be adopted in clinical ambulatory in order to follow-up patients who underwent CABG. This is a well-written, informative and comprehensive review.
Thank you for appreciating this work.
- A figure about the anatomy of coronary arteries would be useful.
Thank you for the suggestion. Indeed, we included in the graphical abstract the imaging of the coronary artery tree in relation to the position of the LIMA and the application of the sound beam. We updated the graphical abstract in order to make the right coronary artery and its branches visible.
- An algorithm that shows the position of DU in the diagnostic process would be informative.
This is a really good point. We included a flow-chart for describing possible suggestions for clinical application of DU in LIMA patency and optimal management of patients who had undergone CABG.
- A section about the limitations and insufficiencies of DU should be added.
Thank for the suggestion. We included the limitations of DU in a dedicated paragraph-
Reviewer 2 Report
I think that this is an interesting review of the use of doppler ultrasound for the assessment of the patency of IMA as a coronary graft for coronary artery disease.
I would wish if authors would expand more on the procedural aspects of this measurements and if they could provide more "how to" and "practical tips" for the use of this method. I think that such addition would be useful to clinicians.
Similarly, are there any studies that compared LIMA or RIMA patency after CABG between doppler ultrasound assessment and CT angiography as the gold standard? I think that this would be useful to add to the text.
Author Response
Reviewer #2
We thank this Reviewer for her/his useful suggestions. We sincerely appreciate his/her comments on our work. This is our point-to-point reply:
- I think that this is an interesting review of the use of doppler ultrasound for the assessment of the patency of IMA as a coronary graft for coronary artery disease.
Thank you for the appreciation of our work.
- I would wish if authors would expand more on the procedural aspects of this measurements and if they could provide more "how to" and "practical tips" for the use of this method. I think that such addition would be useful to clinicians.
We would like to really thank the reviewer for this insight. We included a flow-chart as a possible suggestion for daily use of DU for LIMA patency in daily clinical practice. We also create a dedicated paragraph dealing with this subject as well as with the limitations of this technique.
- Similarly, are there any studies that compared LIMA or RIMA patency after CABG between Doppler ultrasound assessment and CT angiography as the gold standard? I think that this would be useful to add to the text.
This is a really good insight. Indeed, there is no study dealing with the comparison between duplex ultrasound and CT angiography for the evaluation of the patency of LIMA or RIMA in patients who had undergone CABG. We included this point in a dedicated limitation section with the aim to promote studies on this subjects in the next future.